# Double Quantization for Communication-Efficient Distributed Optimization

**Yue Yu**
IIIS, Tsinghua University
yu-y14@mails.tsinghua.edu.cn

**Jiaxiang Wu**
Tencent AI Lab
jonathanwu@tencent.com

**Longbo Huang**[*]
IIIS, Tsinghua University
longbohuang@tsinghua.edu.cn

## Abstract

Modern distributed training of machine learning models often suffers from high communication overhead for synchronizing stochastic gradients and model parameters. In this paper, to reduce the communication complexity, we propose *double quantization*, a general scheme for quantizing both model parameters and gradients. Three communication-efficient algorithms are proposed based on this general scheme. Specifically, (i) we propose a low-precision algorithm AsyLPG with asynchronous parallelism, (ii) we explore integrating gradient sparsification with double quantization and develop Sparse-AsyLPG, (iii) we show that double quantization can be accelerated by the momentum technique and design accelerated AsyLPG. We establish rigorous performance guarantees for the algorithms, and conduct experiments on a multi-server test-bed with real-world datasets to demonstrate that our algorithms can effectively save transmitted bits without performance degradation, and significantly outperform existing methods with either model parameter or gradient quantization.

## 1 Introduction

The *data parallel* mechanism is a widely used architecture for distributed optimization, which has received much recent attention due to data explosion and increasing model complexity. It decomposes the time consuming gradient computations into sub-tasks, and assigns them to separate worker machines for execution. Specifically, the training data is distributed among $M$ workers and each worker maintains a local copy of model parameters. At each iteration, each worker computes a gradient from a mini-batch randomly drawn from its local data. The global stochastic gradient is then computed by synchronously aggregating $M$ local gradients. Model parameters are then updated accordingly.

Two issues significantly slow down methods based on the data parallel architecture. One is the communication cost. For example, all workers must send their entire local gradients to the master node. If gradients are dense, the master node has to receive and send $M \times d$ floating-point numbers per iteration ($d$ is the size of the model vector), which scales linearly with network and model vector sizes. With the increasing computing cluster size and model complexity, it has been observed in many systems that such communication overhead has become the performance bottleneck [38, 35]. The other factor is the synchronization cost, i.e., the master node has to wait for the last local gradient arrival at each iteration. This coordination dramatically increases system's idle time.

---

[*]Corresponding author.

To overcome the first issue, many works focus on reducing the communication complexity of gradients in the data parallel architecture. Generally, there are two approaches. One is quantization [2], which stores gradients using fewer number of bits (lower precision). The other is sparsification [1], i.e., dropping out some coordinates of gradients following certain rules. However, existing communication-efficient algorithms based on data-parallel network still suffer from significant synchronization cost. To address the second issue, many asynchronous algorithms have recently been developed for distributed training [17, 23]. By allowing workers to communicate with master without synchronization, they can effectively improve the training efficiency. Unfortunately, the communication bottleneck caused by transmitting gradients in floating-point numbers still exists.

In this paper, we are interested in jointly achieving communication-efficiency and asynchronous parallelism. Specifically, we study the following composite problem

$$\min_{x \in \Omega} P(x) = f(x) + h(x), \quad f(x) = \frac{1}{n} \sum_{i=1}^{n} f_i(x), \tag{1}$$

where $x \in \mathbb{R}^d$ is the model vector, $f_i(x)$ is smooth, and $h(x)$ is convex but can be nonsmooth. Domain $\Omega \subseteq \mathbb{R}^d$ is a convex set. This formulation has found applications in many different areas, such as operations research, statistics and machine learning [10, 14], e.g., classification or regression. In these problems, $n$ often denotes the number of samples, and $f_i(x)$ denotes the loss function for sample $i$, and $h(x)$ represents certain regularizer.

To solve (1), we propose a novel *double quantization* scheme, which quantizes both model parameters and gradients. This quantization is nontrivial, because we have to deal with low-precision gradients, evaluated on low-precision model vectors. Three communication-efficient algorithms are then proposed under double quantization. We analyze the precision loss of low-precision gradients and prove that these algorithms achieve fast convergence rates while significantly reducing the communication cost. The main contributions are summarized as follows.

(i) We propose an **asy**nchronous **l**ow-**p**recision algorithm AsyLPG to solve the nonconvex and nonsmooth problem (1). We show that AsyLPG achieves the same asymptotic convergence rate as the unquantized serial counterpart, but with a significantly lower communication cost. (ii) We combine gradient sparsification with double quantization and propose Sparse-AsyLPG to further reduce communication overhead. Our analysis shows that the convergence rate scales with $\sqrt{d/\varphi}$ for a sparsity budget $\varphi$. (iii) We propose accelerated AsyLPG, and mathematically prove that double quantization can be accelerated by the momentum technique [19, 26]. (iv) We conduct experiments on a multi-server distributed test-bed. The results validate the efficiency of our algorithms.

## 2   Related Work

Designing large-scale distributed algorithms for machine learning has been receiving increasing attention, and many algorithms, both synchronous and asynchronous, have been proposed, e.g., [22, 4, 17, 12]. In order to reduce the communication cost, researchers also started to focus on cutting down transmitted bits per iteration, based mainly on two schemes, i.e., quantization and sparsification.

**Quantization.** Algorithms based on quantization store a floating-point number using limited number of bits. For example, [25] quantized gradients to a representation of $\{-1, 1\}$, and empirically showed the communication-efficiency in training of deep neural networks. [5, 6] considered the bi-direction communications of gradients between master and workers. In their setting, each worker transmitted gradient sign to the master and master aggregated signs by majority vote. [2, 34, 35] adopted an unbiased gradient quantization with multiple levels. [13] provided a convergence rate of $O(1/\sqrt{K})$ for implementing SGD with unbiased gradient quantizer in solving nonconvex objectives, where $K$ is the number of iterations. The error-feedback method was applied in [25, 35, 29] to integrate history quantization error into the current stage. Specifically, [29] compressed transmitted gradients with error-compensation in both directions between master and workers, and showed a linear speedup in the nonconvex case. [15] constructed several examples where simply transmitting gradient sign cannot converge. They combine the error-feedback method to fix the divergence and prove the convergence rate for nonconvex smooth objectives. [40] also studied bi-direction compression with error-feedback. They partitioned gradients into several blocks, which were compressed using different 1-bit quantizers separately. They analyzed the convergence rate when integrating the momentum. [9] proposed a low-precision framework of SVRG [14], which quantized model parameters for single

machine computation. [38] proposed an end-to-end low-precision scheme, which quantized data, model and gradient with synchronous parallelism. A biased quantization with gradient clipping was analyzed in [37]. [8] empirically studied asynchronous and low-precision SGD on logistic regression. [28] considered the decentralized training and proposed an extrapolation compression method to obtain a higher compression level. [36] proposed a two-phase parameter quantization method, where the parameter in the first phase was the linear combination of full-precision and low-precision parameters. In the second phase, they set the weight of full-precision value to zero to obtain a full compression.

**Sparsification.** Methods on sparsification drop out certain coordinates of transmitted vectors. [1] only transmitted gradients exceeding a threshold. [33, 30] formulated gradient sparsification into an optimization problem to balance sparsity and variance. [31] reduced transmissions in the parameter-server setting by solving a shifted $L_1$ regularized minimization problem. [13] studied a variant of distributed SGD, i.e., only transmitting a subset of model parameters in each iteration. They showed a convergence rate of $O(1/\sqrt{K})$ and a linear speedup as long as all of the model parameters were transmitted in limited consecutive iterations. Recently, [27, 3] analyzed the convergence behavior of sparsified SGD with memory, i.e., compensating gradient with sparsification error.

Our work distinguishes itself from the above results in: (i) we quantize both model vectors and gradients, (ii) we integrate gradient sparsification into double quantization and prove convergence, and (iii) we analyze how double quantization can be accelerated to reduce communication rounds.

**Notation.** $x^*$ is the optimal solution of (1). $||x||_\infty$, $||x||_1$ and $||x||$ denote the max, $L_1$ and $L_2$ norms of $x$, respectively. For a vector $v_t \in \mathbb{R}^d$, $[v_t]_i$ or $v_{t,i}$ denotes its $i$-th coordinate. $\{e_i\}_{i=1}^d$ is the standard basis in $\mathbb{R}^d$. The base of logarithmic function is 2. $\tilde{O}(f)$ denotes $O(f \cdot polylog(f))$. We use the proximal operator to handle a nonsmooth function $h(x)$, i.e., $\text{prox}_{\eta h}(x) = \arg\min_y h(y) + \frac{1}{2\eta}||y - x||^2$. If problem (1) is nonconvex and nonsmooth, we apply the commonly used convergence metric *gradient mapping* [20], i.e., $G_\eta(x) \triangleq \frac{1}{\eta}[x - \text{prox}_{\eta h}(x - \eta \nabla f(x))]$. $x$ is defined as an $\epsilon$-accurate solution if it satisfies $\mathbf{E}||G_\eta(x)||^2 \le \epsilon$.

# 3 Preliminary

**Low-Precision Representation via Quantization.** Low-precision representation stores numbers using limited number of bits, contrast to the 32-bit full-precision.[2] It can be represented by a tuple $(\delta, b)$, where $\delta \in \mathbb{R}$ is the scaling factor and $b \in \mathbf{N}^+$ is the number of bits used. Specifically, given a tuple $(\delta, b)$, the set of representable numbers is given by

$$\text{dom}(\delta, b) = \{-2^{b-1} \cdot \delta, ..., -\delta, 0, \delta, ..., (2^{b-1} - 1) \cdot \delta\}.$$

For any full-precision $x \in \mathbb{R}$, we call the procedure of transforming it to a low-precision representation as quantization, which is denoted by function $Q_{(\delta,b)}(x)$. It outputs a number in $\text{dom}(\delta, b)$ according to the following rules:

(i) If $x$ lies in the convex hull of $\text{dom}(\delta, b)$, i.e., there exists a point $z \in \text{dom}(\delta, b)$ such that $x \in [z, z + \delta]$, then $x$ will be stochastically rounded in an unbiased way:

$$Q_{(\delta,b)}(x) = \begin{cases} z + \delta, & \text{with probability } \frac{x-z}{\delta}, \\ z, & \text{with probability } \frac{z+\delta-x}{\delta}. \end{cases}$$

(ii) Otherwise, $Q_{(\delta,b)}(x)$ outputs the closest point to $x$ in $\text{dom}(\delta, b)$.

This quantization method is widely used in existing works, e.g., [38, 2, 34, 9], sometimes under different formulation. In the following sections, we adopt $Q_{(\delta,b)}(v)$ to denote quantization on vector $v \in \mathbb{R}^d$, which means that each coordinate of $v$ is independently quantized using the same tuple $(\delta, b)$. Low-precision representation can effectively reduce communication cost, because we only need $(32 + bd)$ bits to transmit the quantized $Q_{(\delta,b)}(v)$ (32 bits for $\delta$, and $b$ bits for each coordinate), whereas it needs $32d$ bits for a full-precision $v$.

**Algorithm 1** AsyLPG

---

1: **Input:** $S$, $m$, $\eta$, $b_x$, $b$, $\tilde{x}^0 = x^0$;
2: **for** $s = 0, 1, ..., S-1$ **do**
3:     $x_0^{s+1} = \tilde{x}^s$;
4:     Compute $\nabla f(\tilde{x}^s) = \frac{1}{n}\sum_{i=1}^{n}\nabla f_i(\tilde{x}^s)$;    /\* **Map-reduce global gradient computation**
5:     **for** $t = 0$ **to** $m-1$ **do**
6:       /\* **For master:**
7:       (i) **Model Parameter Quantization:** Set $\delta_x = \frac{||x_{D(t)}^{s+1}||_\infty}{2^{b_x-1}-1}$ and quantize $x_{D(t)}^{s+1}$ subject to (2).
          Then, send $Q_{(\delta_x, b_x)}(x_{D(t)}^{s+1})$ to workers;
8:       (ii) Receive local gradient $\zeta_t$, update $u_t^{s+1} = \zeta_t + \nabla f(\tilde{x}^s)$, $x_{t+1}^{s+1} = \text{prox}_{\eta h}(x_t^{s+1} - \eta u_t^{s+1})$;
9:       /\* **For worker:**
10:       (i) Receive $Q_{(\delta_x, b_x)}(x_{D(t)}^{s+1})$, stochastically sample a data-point $a \in \{1, ..., n\}$, and calculate
          gradient $\alpha_t = \nabla f_a(Q_{(\delta_x, b_x)}(x_{D(t)}^{s+1})) - \nabla f_a(\tilde{x}^s)$;
11:       (ii) **Gradient Quantization:** Set $\delta_{\alpha_t} = \frac{||\alpha_t||_\infty}{2^{b-1}-1}$ and send the quantized gradient $\zeta_t = Q_{(\delta_{\alpha_t}, b)}(\alpha_t)$ to the master;
12:     **end for**
13:     $\tilde{x}^{s+1} = x_m^{s+1}$;
14: **end for**
15: **Output:** Uniformly choosing from $\{\{x_t^{s+1}\}_{t=0}^{m-1}\}_{s=0}^{S-1}$.

---

**Distributed Network with Asynchronous Communication.** As shown in Figure 1, we consider a network with one master and multiple workers, e.g., the parameter-server setting.
The master maintains and updates a model vector $x$, and keeps a training clock. Each worker can get access to the full datasets and keeps a disjoint partition of data. In each communication round, a worker retrieves $x$ from the master, evaluates the gradient $g(x)$, and then sends it back to the master. Since workers asynchronously pull and push data during the training process, at a time $t$, the master may use a delayed gradient

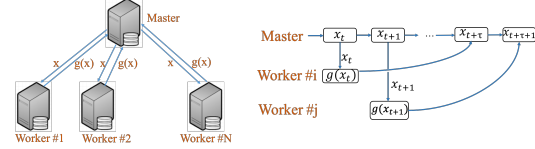

Figure 1: The framework of distributed network with asynchronous communication. Left: network structure. Right: training process.

calculated on a previous $x_{D(t)}$, where $D(t) \leq t$. Many works showed that a near linear speedup can be achieved if the delay is reasonably moderate [17, 23].

## 4 Algorithms

To solve problem (1), we propose a communication-efficient algorithm with double quantization, namely AsyLPG, and introduce its two variants with gradient sparsification and momentum acceleration. We begin with the assumptions made in this paper. They are mild and are often assumed in the literature, e.g., [14, 24].

**Assumption 1.** *The stochastically sampled gradient is unbiased, i.e., for $a \in \{1, ..., n\}$ sampled in Algorithms 1, 2, 3, $\mathbf{E}_a[\nabla f_a(x)] = \nabla f(x)$. Moreover, the random variables in different iterations are independent.*

**Assumption 2.** *Each $f_i(x)$ in (1) is $L$-smooth, i.e., $||\nabla f_i(x) - \nabla f_i(y)|| \leq L||x - y||$, $\forall x, y \in \Omega$.*

**Assumption 3.** *The gradient delay is bounded by some finite constant $\tau > 0$, i.e., $t - D(t) \leq \tau$, $\forall t$.*

### 4.1 Communication-Efficient Algorithm with Double Quantization: AsyLPG

In this section, we introduce our new distributed algorithm AsyLPG, with **asy**nchronous communication and **l**ow-**p**recision floating-point representation. As shown in Algorithm 1, AsyLPG divides the training procedure into epochs, similar to SVRG [14], with each epoch containing $m$ inner iterations. At the beginning of each epoch, AsyLPG performs one round of communication between the master and workers to calculate the full-batch gradient $\nabla f(\tilde{x}^s)$, where $\tilde{x}^s$ is a snapshot variable evaluated at

the end of each epoch $s$. This one round communication involves full-precision operation because it is only performed once per epoch, and its communication overhead is small compared to the subsequent $m$ communication rounds in inner iterations, where the model parameters are updated.

In inner iterations, the communication between the master and workers utilizes the asynchronous parallelism described in Section 3. To reduce communication complexity, we propose *double quantization*, i.e., quantizing both model parameters and gradients.

**Model Parameter Quantization.** Prior works [9, 39] showed that simply quantizing model vectors using a constant tuple $(\delta, b)$ fails to converge to the optimal solution due to a non-diminishing quantization error. In our case, we impose the additional requirement that the chosen $(\delta, b)$ satisfies the following condition (see Step 7 of AsyLPG):

$$\mathbf{E}_Q||Q_{(\delta_x, b_x)}(x_{D(t)}^{s+1}) - x_{D(t)}^{s+1}||^2 \le \mu||x_{D(t)}^{s+1} - \tilde{x}^s||^2. \tag{2}$$

This condition is set to control the precision loss of $x_{D(t)}^{s+1}$ with a dynamic value of $||x_{D(t)}^{s+1} - \tilde{x}^s||^2$ and a positive hyperparameter $\mu$, so as to achieve an accurate solution. Note that with a larger $\mu$, we can aggressively save more transmitted bits (using a smaller $b_x$). In practice, the precision loss and communication cost can be balanced by selecting a proper $\mu$. On the other hand, from

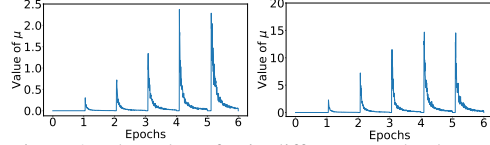

Figure 2: The value of $\mu$ in different epochs that guarantees (2). Left: $b_x = 8$. Right: $b_x = 4$. The statistics are based on a logistic regression on dataset *covtype* [7].

the analysis aspect, we can always find a $\mu$ that guarantees (2) throughout the training process, for any given $b_x$. In the special case when $x_{D(t)}^{s+1} = \tilde{x}^s$, the master only needs to send a flag bit since $\tilde{x}^s$ has already been stored at the workers, and (2) still holds. Figure 2 validates the practicability of (2), where we plot the value of $\mu$ required to guarantee (2) given $b_x$. Note that the algorithm already converges in both graphs. In this case, we see that when $b_x$ is 4 or 8, $\mu$ can be upper bounded by a constant. Also, by setting a larger $\mu$, we can choose a smaller $b_x$. The reason $\mu$ increases at the beginning of each epoch is because $||x_{D(t)}^{s+1} - \tilde{x}^s||^2$ is small. After several inner iterations, $x_{D(t)}^{s+1}$ moves further away from $\tilde{x}^s$. Thus, a smaller $\mu$ suffices to guarantee (2).

**Gradient Quantization.** After receiving the low-precision model parameter, as shown in Steps 10-11, a worker calculates a gradient $\alpha_t$ and quantizes it into its low-precision representation $Q_{(\delta_{\alpha_t}, b)}(\alpha_t)$, and then sends it to the master.

In Step 8, the master constructs a semi-stochastic gradient $u_t^{s+1}$ based on the received low-precision $Q_{(\delta_{\alpha_t}, b)}(\alpha_t)$ and the full-batch gradient $\nabla f(\tilde{x}^s)$, and updates the model vector $x$ using step size $\eta$. The semi-stochastic gradient evaluated here adopts the variance reduction method proposed in SVRG [14] and is used to accelerate convergence. If Algorithm 1 is run without *double quantization* and asynchronous parallelism, i.e., only one compute node with no delay, [24] showed that:

**Theorem 1.** *([24], Theorem 5) Suppose $h(x)$ is convex and Assumptions 1, 2 hold. Let $T = Sm$ and $\eta = \rho/L$ where $\rho \in (0, 1/2)$ and satisfies $4\rho^2 m^2 + \rho \le 1$. Then for the output $x_{out}$ of Algorithm 1, we have $\mathbf{E}||G_\eta(x_{out})||^2 \le \big(2L(P(x^0) - P(x^*))\big)/\big(\rho(1 - 2\rho)T\big).$*

### 4.1.1 Theoretical Analysis

**Lemma 1.** *Denote $\Delta = \frac{d}{4(2^{b-1}-1)^2}$. If Assumptions 1, 2, 3 hold, then for the gradient $u_t^{s+1}$ in Algorithm 1, its variance can be bounded by*

$$\mathbf{E}||u_t^{s+1} - \nabla f(x_t^{s+1})||^2 \le 2L^2(\mu + 1)(\Delta + 2)\mathbf{E}\Big[||x_{D(t)}^{s+1} - x_t^{s+1}||^2 + ||x_t^{s+1} - \tilde{x}^s||^2\Big].$$

**Theorem 2.** *Suppose $h(x)$ is convex, conditions in Lemma 1 hold, $T = Sm$, $\eta = \frac{\rho}{L}$, where $\rho \in (0, \frac{1}{2})$, and $\rho, \tau$ satisfy $8\rho^2 m^2(\mu + 1)(\Delta + 2) + 2\rho^2(\mu + 1)(\Delta + 2)\tau^2 + \rho \le 1$. Then, for the output $x_{out}$ of Algorithm 1 , we have $\mathbf{E}||G_\eta(x_{out})||^2 \le \big(2L(P(x^0) - P(x^*))\big)/\big(\rho(1 - 2\rho)T\big).$*

**Remarks.** From Theorem 2, we see that if $\mu = O(1)$, $b = O(\log\sqrt{d})$ and $\rho = O(\frac{1}{m})$, AsyLPG achieves the same asymptotic convergence rate as in Theorem 1, while transmitting much fewer bits ($b = O(\log\sqrt{d})$ is much smaller than 32). Our analytical results focus on convergence, as done in [35, 34]. Exactly quantifying the amount of improvement on communication complexity, however, remains challenging due to the complicated dynamics of parameter updates, e.g., [9, 28].

---

**Algorithm 2** Sparse-AsyLPG: Procedures for worker

---

1: (i) Receive $Q_{(\delta_x,b_x)}(x_{D(t)}^{s+1})$, stochastically sample a data-point $a \in \{1,...,n\}$, and calculate gradient $\alpha_t = \nabla f_a(Q_{(\delta_x,b_x)}(x_{D(t)}^{s+1})) - \nabla f_a(\tilde{x}^s)$;

2: (ii) **Gradient Sparsification:** Select a budget $\varphi_t$ and sparsify $\alpha_t$ to obtain $\beta_t$ using (3);

3: (iii) **Gradient Quantization:** Set $\delta_{\beta_t} = \frac{||\beta_t||_\infty}{2^{b-1}-1}$, and send $\zeta_t = Q_{(\delta_{\beta_t},b)}(\beta_t)$ to the master;

---

We instead show through extensive experiments that our scheme significantly improves upon existing benchmarks, e.g., Figure 3 and Table 1. Note that AsyLPG can also adopt other gradient quantization methods (even biased ones), e.g., [37], and similar results can be established.

## 4.2 AsyLPG with Gradient Sparsification

In this section, we explore how to further reduce the communication cost by incorporating gradient sparsification into double quantization, and propose a new algorithm Sparse-AsyLPG. As shown in Algorithm 2, after calculating $\alpha_t$, workers successively perform sparsification and quantization on it. Specifically, we drop out certain coordinates of $\alpha_t$ to obtain a sparsified vector $\beta_t$ according to the following rules [33]:

$$\beta_t = \left[ Z_1 \frac{\alpha_{t,1}}{p_1}, Z_2 \frac{\alpha_{t,2}}{p_2}, ..., Z_d \frac{\alpha_{t,d}}{p_d} \right], \tag{3}$$

where $Z = [Z_1, Z_2, ..., Z_d]$ is a binary-valued vector with $Z_i \sim \text{Bernoulli}(p_i)$, $0 < p_i \leq 1$, and $Z_i$'s are independent. Thus, $\beta_t$ is obtained by randomly selecting the $i$-th coordinate of $\alpha_t$ with probability $p_i$. It can be verified that $\mathbf{E}[\beta_t] = \alpha_t$. Define $\varphi_t \triangleq \sum_{i=1}^d p_i$ to measure the sparsity of $\beta_t$. To reduce the communication complexity, it is desirable to make $\varphi_t$ as small as possible, which, on the other hand, brings about a large variance. The following lemma quantifies the relationship between $\beta_t$ and $\varphi_t$, and is derived based on results in [30].

**Lemma 2.** *Suppose $\varphi_t \leq \frac{||\alpha_t||_1}{||\alpha_t||_\infty}$. Then, for $\alpha_t = \sum_{i=1}^d \alpha_{t,i} e_i$ and $\beta_t$ generated in (3), we have* $\mathbf{E}||\beta_t||^2 \geq \frac{1}{\varphi_t}||\alpha_t||_1^2$. *The equality holds if and only if $p_i = \frac{|\alpha_{t,i}| \cdot \varphi_t}{||\alpha_t||_1}$.*

Based on Lemma 2, in Step 2 of Algorithm 2, we can select a sparsity budget $\varphi_t \leq \frac{||\alpha_t||_1}{||\alpha_t||_\infty}$ and set $p_i = \frac{|\alpha_{t,i}| \cdot \varphi_t}{||\alpha_t||_1}$ to minimize the variance of $\beta_t$. Then, we quantize $\beta_t$ and send its low-precision version to the master. The master node in Sparse-AsyLPG employs the same model parameter quantization and updates parameter $x$ in the same manner as in AsyLPG. The asynchronous parallelism is also applied in the communications between master and workers.

### 4.2.1 Theoretical Analysis

We first study the variance of the sparsified gradient $u_t^{s+1} = \zeta_t + \nabla f(\tilde{x}^s)$.

**Lemma 3.** *Suppose $\varphi_t \leq \frac{||\alpha_t||_1}{||\alpha_t||_\infty}$, Assumptions 1, 2, 3 hold, and for each $i \in \{1,...,d\}$, $p_i = \frac{|\alpha_{t,i}| \cdot \varphi_t}{||\alpha_t||_1}$. Denote $\Gamma = \frac{d^2}{4\varphi(2^{b-1}-1)^2} + \frac{d}{\varphi} + 1$, where $\varphi = \min_t\{\varphi_t\}$. Then, for the gradient $u_t^{s+1}$ in Sparse-AsyLPG, we have $\mathbf{E}||u_t^{s+1} - \nabla f(x_t^{s+1})||^2 \leq 2L^2(\mu+1)\Gamma \mathbf{E}\left[||x_{D(t)}^{s+1} - x_t^{s+1}||^2 + ||x_t^{s+1} - \tilde{x}^s||^2\right]$.*

**Theorem 3.** *Suppose $h(x)$ is convex, conditions in Lemma 3 hold, $T = Sm$, $\eta = \frac{\rho}{L}$, where $\rho \in (0, \frac{1}{2})$, and $\rho, \tau$ satisfy $8\rho^2 m^2(\mu+1)\Gamma + 2\rho^2(\mu+1)\tau^2\Gamma + \rho \leq 1$. Then, for the output $x_{out}$ of Sparse-AsyLPG, we have $\mathbf{E}||G_\eta(x_{out})||^2 \leq \left(2L(P(x^0) - P(x^*))\right)/\left(\rho(1-2\rho)T\right)$.*

**Remarks.** Setting $b = O(\log\sqrt{d})$, we obtain $\Gamma = O(d/\varphi)$. We then conclude from Theorem 3 that Sparse-AsyLPG converges with a rate linearly scales with $\sqrt{d/\varphi}$, and significantly reduces transmitted bits per iteration. Note that before transmitting $\zeta_t$, we need to encode it to a string, which contains 32 bits for $\delta_{\beta_t}$ and $b$ bits for each coordinate. Since $\beta_t$ is sparse, we only encode the nonzero coordinates, i.e., using $\log d$ bits to encode the position of a nonzero element followed by its value.

---
**Algorithm 3** Acc-AsyLPG
---
1: **Input:** $S, m, b_x, b, \tilde{x}^0, y_m^0 = \tilde{x}^0$;
2: **for** $s = 1, 2, ..., S$ **do**
3:     update $\theta_s, \eta_s, x_0^s = \theta_s y_0^s + (1 - \theta_s)\tilde{x}^{s-1}, y_0^s = y_m^{s-1}$;
4:     Compute $\nabla f(\tilde{x}^{s-1}) = \frac{1}{n}\sum_{i=1}^n \nabla f_i(\tilde{x}^{s-1})$; /\* **Map-reduce global gradient computation**
5:     **for** $t = 0$ **to** $m - 1$ **do**
6:         /\* **For master:**
7:         (i) **Model Parameter Quantization:** Set $\delta_x = \frac{||x_{D(t)}^s||_\infty}{2^{b_x-1}-1}$, and quantize $x_{D(t)}^s$ subject to (4).
             Then, send $Q_{(\delta_x, b_x)}(x_{D(t)}^s)$ to workers;
8:         (ii) **Momentum Acceleration:**
             Receive local gradient $Q_{(\delta_{\alpha_t}, b)}(\alpha_t)$, compute $u_t^s = Q_{(\delta_{\alpha_t}, b)}(\alpha_t) + \nabla f(\tilde{x}^{s-1})$ and update
             $y_{t+1}^s = \text{prox}_{\eta_s h}(y_t^s - \eta_s u_t^s)$, $x_{t+1}^s = \tilde{x}^{s-1} + \theta_s(y_{t+1}^s - \tilde{x}^{s-1})$;
9:         /\* **For worker:**
10:        (i) Receive $Q_{(\delta_x, b_x)}(x_{D(t)}^s)$, stochastically sample a data-point $a \in \{1, ..., n\}$ and calculate
             gradient $\alpha_t = \nabla f_a(Q_{(\delta_x, b_x)}(x_{D(t)}^s)) - \nabla f_a(\tilde{x}^{s-1})$;
11:        (ii) **Gradient Quantization:** quantize $\alpha_t$ using Step 11 in Algorithm 1;
12:     **end for**
13:     $\tilde{x}^s = \frac{1}{m}\sum_{t=0}^{m-1} x_{t+1}^s$;
14: **end for**
15: **Output:** $\tilde{x}^S$.
---

### 4.3 Accelerated AsyLPG

In the above, we mainly focus on reducing the communication cost within each iteration. Here we propose an algorithm with an even faster convergence and fewer communication rounds. Specifically, we incorporate the popular momentum or Nesterov technique [19, 26] into AsyLPG. To simplify presentation, we only present accelerated AsyLPG (Acc-AsyLPG) in Algorithm 3. The method can similarly be applied to Sparse-AsyLPG.

Algorithm 3 still adopts asynchronous parallelism and double quantization, and makes the following key modifications. (i) In Step 7, the model parameter quantization satisfies

$$\mathbf{E}_Q||Q_{(\delta_x, b_x)}(x_{D(t)}^s) - x_{D(t)}^s||^2 \leq \theta_s\mu||x_{D(t)}^s - \tilde{x}^{s-1}||^2, \tag{4}$$

where $\mu$ is the hyperparameter that controls the precision loss. $\theta_s$ is the momentum weights and its value will be specified later. (ii) Momentum acceleration is implemented in Steps 3 and 8, through an auxiliary variable $y_{t+1}^s$. The update of $x_{t+1}^s$ combines history information $\tilde{x}^{s-1}$ and $y_{t+1}^s$. In the following, we show that with the above modifications, Acc-AsyLPG achieves an even faster convergence rate.

**Theorem 4.** *Suppose each $f_i(x)$ and $h(x)$ are convex, Assumptions 1, 2, 3 hold, and the domain $\Omega$ of $x$ is bounded by D, such that $\forall x, y \in \Omega, ||x - y||^2 \leq D$. Let $\theta_s = \frac{2}{s+2}$, $\eta_s = \frac{1}{\sigma L \theta_s}$, where $\sigma > 1$ is a constant. If $\sigma, \tau$ satisfy $\tau \leq \frac{1}{2}\left[\sqrt{\left(\frac{2}{\gamma\theta_s} + \theta_s\Delta\right)^2 + \frac{4(\sigma-1)}{\gamma}} - \left(\frac{2}{\gamma\theta_s} + \theta_s\Delta\right)\right]$ where $\Delta = \frac{d}{(2^{b-1}-1)^2} + 2$, $\gamma = 1 + 2\theta_s\mu$, then under Algorithm 3, we have $\mathbf{E}[P(\tilde{x}^S) - P(x^*)] \leq \tilde{O}((L/m + LD\mu\Delta/\tau + LD\mu)/S^2)$.*

The bounded domain condition in Theorem 4 is commonly assumed in literature, e.g., [32], and the possibility of going outside domain is avoided by the proximal operator in Step 8. If $b = O(\log\sqrt{d})$ and $\mu = O(1)$, the constraint of delay $\tau$ can be easily satisfied with a moderate $\sigma$. Then, our Acc-AsyLPG achieves acceleration while effectively reducing the communication cost.

## 5 Experiments

We conduct experiments to validate the efficiency of our algorithms. We start with the logistic regression problem and then evaluate the performance of our algorithms on neural network models. We further study the relationship of hyperparameter $\mu$ and number of transmitted bits.

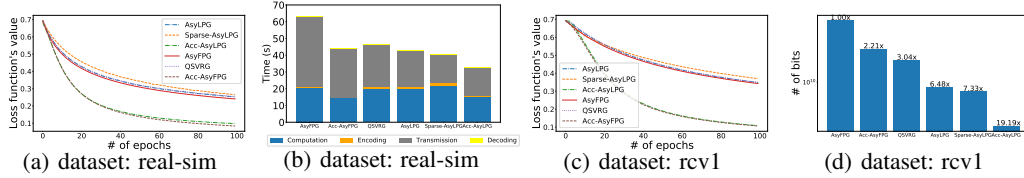

(a) dataset: real-sim      (b) dataset: real-sim      (c) dataset: rcv1      (d) dataset: rcv1

Figure 3: (a) and (c): training curves on *real-sim* and *rcv1*. (b): decomposition of time consumption (recorded until the training loss is first below 0.5). (d): # of transmitted bits until the training loss is first below 0.4.

## 5.1 Evaluations on Logistic Regression.

We begin with logistic regression on dataset *real-sim* [7]. The evaluations are setup on a 6-server distributed test-bed. Each server has 16 cores and 16GB memory. The communication among servers is handled by OpenMPI [21]. We use $L_1$, $L_2$ regularization with weights $10^{-5}$ and $10^{-4}$, respectively. The mini-batch size $B = 200$ and epoch length $m = \lceil \frac{n}{B} \rceil$. The following six algorithms are compared, using a constant learning rate (denoted as *lr*) tuned to achieve the best result from $\{1e^{-1}, 1e^{-2}, 5e^{-2}, 1e^{-3}, 5e^{-3}, ..., 1e^{-5}, 5e^{-5}\}$.

(i) AsyLPG, Sparse-AsyLPG, Acc-AsyLPG. We set $b_x = 8$ and $b = 8$ in these three algorithms. The sparsity budget in Sparse-AsyLPG is selected as $\varphi_t = ||\alpha_t||_1/||\alpha_t||_\infty$. We do not tune $\varphi_t$ to present a fair comparison. Parameters in Acc-AsyLPG are set to be $\theta_s = 2/(s + 2)$ and $\eta_s = lr/\theta_s$.

(ii) QSVRG [2], which is a *gradient-quantized* algorithm. We implement it in an asynchronous-parallelism way. Its gradient quantization method is equivalent to Step 11 in AsyLPG. If run with synchronization and without quantization, QSVRG and AsyLPG have the same convergence rate. For a fair comparison, we set the gradient quantization bit $b = 8$ for QSVRG.

(iii) The full-precision implementations of AsyLPG and Acc-AsyLPG, denoted as AsyFPG and Acc-AsyFPG, respectively. In both algorithms, we remove double quantization.

**Convergence and time consumption.** Figure 3 presents the evaluations on dataset *real-sim*. The plot (a) shows that AsyLPG and Acc-AsyLPG have similar convergence rates to their full-precision counterparts. Our Sparse-AsyLPG also converges fast with a very small accuracy degradation. The time consumption presented in the plot (b) shows the communication-efficiency of our algorithms. With similar convergence rates, our low-precision algorithms significantly reduce the communication overhead when achieving the same training loss. Moreover, the comparison between AsyLPG and QSVRG validates the redundancy of 32 bits representation of model parameter.

**Communication complexity.** We experimented logistic regression on dataset *rcv1* [7]. The $L_1$ and $L_2$ regularization are adopted, both with weights $10^{-4}$. *lr* is tuned in the same way as *real-sim*. In Figure 3(d), we record the total number of transmitted bits. It shows that AsyLPG, Sparse-AsyLPG, Acc-AsyLPG can save up to $6.48\times$, $7.33\times$ and $19.19\times$ bits compared to AsyFPG.

## 5.2 Evaluations on Neural Network

We conduct evaluations on dataset MNIST [18] using a 3-layer fully connected neural network. The 6-server distributed test-bed in Section 5.1 is adopted. The hidden layer contains 100 nodes, and uses ReLU activation function. Softmax loss function and $L_2$ regularizer with weight $10^{-4}$ are adopted. We use 10k training and 2k test samples which are randomly drawn from the full dataset (60k training / 10k test). The mini-batch size is 20 and the epoch size $m$ is 500. We set $b_x = 8$ and $b = 4$ for low-precision algorithms.

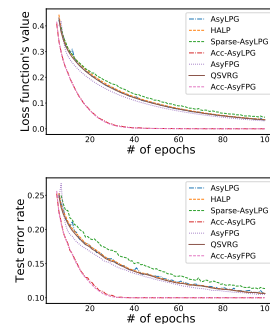

Figure 4: Evaluation on dataset MNIST. Top: the training curve. Bottom: the test error rate.

We also compare our algorithms with HALP [9], which is a low-precision variant of SVRG with *quantized model vectors*. We implement its distributed version with asynchronous parallelism. For a fair comparison, we also set $b_x = 8$ for HALP. *lr* is constant and is tuned to achieve the best result for each algorithm. Figure 4 presents the convergence of the seven algorithms. In the left table of Table 1, we record the total number of transmitted bits. We see that the results are similar as in the logistic regression case, i.e., our new low-precision algorithms can significantly reduce communication overhead compared to their full-precision counterparts and

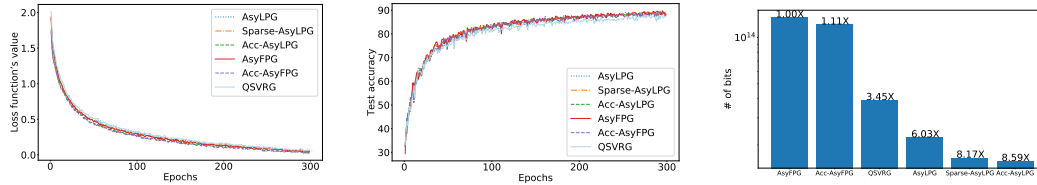

Figure 6: Evaluations on CIFAR10: training loss (1st column), test accuracy (2nd column) and total number of transmitted bits.

Table 1: Evaluation on dataset MNIST. Left: # of transmitted bits until the training loss is first below $0.05$. Right: The value of $b_x$ and # of transmitted bits of AsyLPG under different $\mu$.

| Algorithm | # bits | Ratio |
|---|---|---|
| AsyFPG | $2.42e9$ | – |
| Acc-AsyFPG | $6.87e8$ | $3.52\times$ |
| QSVRG | $4.50e8$ | $5.38\times$ |
| HALP | $7.36e8$ | $3.29\times$ |
| AsyLPG | $3.33e8$ | $7.28\times$ |
| Sparse-AsyLPG | $2.73e8$ | $8.87\times$ |
| Acc-AsyLPG | $1.26e8$ | $19.13\times$ |

| $\mu$ | $b_x$ | # bits | $\mu$ | $b_x$ | # bits |
|---|---|---|---|---|---|
| 0.005 | 11 | $2.42e7$ | 2.0 | 6 | $1.24e7$ |
| 0.01 | 10 | $2.33e7$ | 10 | 5 | $1.14e7$ |
| 0.05 | 9 | $1.52e7$ | 50 | 4 | $1.08e7$ |
| 0.1 | 8 | $1.07e7$ | 150 | 3 | $1.44e7$ |
| 0.5 | 7 | $1.12e7$ | 800 | 2 | $2.16e8$ |

QSVRG. Moreover, our AsyLPG with double quantization has comparable convergence rate to HALP while sending much less bits.

**Study of $\mu$.** The hyperparameter $\mu$ is set to control the precision loss incurred by model parameter quantization. Before, we fix $b_x$ to compare our algorithms with other methods. Now we study the relationship of $\mu$ and transmitted bits. Note that when $\mu$ is fixed, we choose $b_x$ to satisfy (2). In Figure 5, we set $\mu = 0.6$ and study how the accuracy of model quantizer improves with iterations when running AsyLPG on MNIST. We see that the quantization error diminishes. Thus, the number of transmitted bits increases as the number of iteration grows.

Next, in the right table of Table 1, we study the performance of AsyLPG under different $\mu$. The value $b_x$ is also chosen by guaranteeing (2). We provide the overall numbers of transmitted bits under different $b_x$ until the training loss is first less than $0.5$. The results validate that with the increasing $\mu$, we can choose a smaller $b_x$, to save more communication cost per iteration. The total number of transmitted bits decreases until a threshold $\mu = 0.5$, beyond which significant precision loss happens and we need more training iterations for achieving the same accuracy.

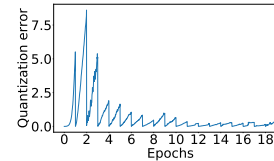

Figure 5: The accuracy of model quantizer.

**Evaluations on Deep Model.** We further set up experiments on PyTorch with ResNet18 [11] on CIFAR10 dataset [16]. The model size is about $44$MB. We use 50k training samples and 10k evaluation samples. For direct comparison, no data augmentation is used. The batch size is $128$. The learning rate starts from $0.1$, and is divided by 10 at 150 and 250 epochs. We set $b_x = 8$, $b = 4$ for low-precision algorithms. The sparsity budget $\varphi_t = ||\alpha_t||_1/||\alpha_t||_\infty$. In Figure 6, we plot the training loss and test accuracy with respect to epochs, and provide the total transmitted bits until the training loss first gets below $0.17$. It shows that our algorithms achieve similar accuracy and effectively reduce the communication cost compared to benchmarks.

# 6 Conclusion

We propose three communication-efficient algorithms for distributed training with asynchronous parallelism. The key idea is quantizing both model parameters and gradients, called double quantization. We analyze the variance of low-precision gradients and show that our algorithms achieve the same asymptotic convergence rate as the full-precision algorithms, while transmitting much fewer bits per iteration. We also incorporate gradient sparsification into double quantization, and setup relation between convergence rate and sparsity budget. We accelerate double quantization by integrating momentum techniques. The evaluations on logistic regression and neural network based on real-world datasets validate that our algorithms can significantly reduce communication cost.

**Acknowledgments**

The work of Yue Yu and Longbo Huang was supported in part by the National Natural Science Foundation of China Grant 61672316, the Zhongguancun Haihua Institute for Frontier Information Technology and the Turing AI Institute of Nanjing. We would like to thank anonymous reviewers for their insightful suggestions.

## Footnotes

[2]We assume without loss of generality that a floating-point number is stored using 32 bits (also see, e.g., [2, 37]). Our results can extend to the case when numbers are stored with other precision.

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
