[Supplementary Material]

# Supplementary Materials for "Double Quantization for Communication-Efficient Distributed Optimization"

## 1 Convergence Analysis for AsyLPG

**Lemma 1.** *For a vector $v \in \mathbb{R}^d$, if $\delta = \frac{||v||_\infty}{2^{b-1}-1}$ or $\frac{||v||_2}{2^{b-1}-1}$, we have*

$$\mathbf{E}||Q_{(\delta,b)}(v) - v||^2 \leq \frac{d\delta^2}{4}. \tag{1}$$

*Proof.* Because the squared $L_2$ norm separates along dimensions and each coordinate of $v$ is independently quantized, we only need to prove $\mathbf{E}||Q_{(\delta,b)}([v]_i) - [v]_i||^2 \leq \frac{\delta^2}{4}$, for all $i \in \{1, ..., d\}$. If the scaling factor $\delta = \frac{||v||_\infty}{2^{b-1}-1}$ or $\frac{||v||_2}{2^{b-1}-1}$, it can be verified that $[v]_i$ locates in the convex hull of $\mathrm{dom}(\delta, b)$ and $Q_{(\delta,b)}(v)$ is an unbiased quantization. Then $\mathbf{E}||Q_{(\delta,b)}([v]_i) - [v]_i||^2 \leq \frac{\delta^2}{4}$ according to Lemma 1 in [4]. $\square$

**Lemma 2.** *If Assumptions $1, 2, 3$ hold, then for the gradient $u_t^{s+1}$ in Algorithm $1$, its variance can be bounded by*

$$\mathbf{E}||u_t^{s+1} - \nabla f(x_t^{s+1})||^2 \leq 2L^2(\mu+1)(\Delta+2)\mathbf{E}\Big[||x_{D(t)}^{s+1} - x_t^{s+1}||^2 + ||x_t^{s+1} - \tilde{x}^s||^2\Big], \tag{2}$$

*where $\Delta = \frac{d}{4(2^{b-1}-1)^2}$.*

*Proof.*

$$
\begin{aligned}
&\mathbf{E}||u_t^{s+1} - \nabla f(x_t^{s+1})||^2 \\
&= \mathbf{E}||Q_{(\delta_{\alpha_t},b)}(\alpha_t) + \nabla f(\tilde{x}^s) - \nabla f(x_t^{s+1})||^2 \\
&= \mathbf{E}||Q_{(\delta_{\alpha_t},b)}(\alpha_t) - \alpha_t + \alpha_t + \nabla f(\tilde{x}^s) - \nabla f(x_t^{s+1})||^2 \\
&= \mathbf{E}||Q_{(\delta_{\alpha_t},b)}(\alpha_t) - \alpha_t||^2 + \mathbf{E}||\alpha_t + \nabla f(\tilde{x}^s) - \nabla f(x_t^{s+1})||^2 \\
&\leq \Delta \underbrace{\mathbf{E}||\alpha_t||^2}_{T_1} + \underbrace{\mathbf{E}||\alpha_t + \nabla f(\tilde{x}^s) - \nabla f(x_t^{s+1})||^2}_{T_2},
\end{aligned} \tag{3}
$$

where the third equality holds because $\delta_{\alpha_t} = \frac{||\alpha_t||_\infty}{2^{b-1}-1}$ and $Q_{(\delta_{\alpha_t},b)}(\alpha_t)$ is an unbiased quantization. The final inequality follows from Lemma 1. Next we bound $T_1$ and $T_2$.

$$
\begin{aligned}
T_1 &= \mathbf{E}||\nabla f_a(Q_{(\delta_x,b_x)}(x_{D(t)}^{s+1})) - \nabla f_a(\tilde{x}^s)||^2 \\
&\leq L^2 \mathbf{E}||Q_{(\delta_x,b_x)}(x_{D(t)}^{s+1}) - \tilde{x}^s||^2 \\
&= L^2 \mathbf{E}||Q_{(\delta_x,b_x)}(x_{D(t)}^{s+1}) - x_{D(t)}^{s+1}||^2 + L^2 \mathbf{E}||x_{D(t)}^{s+1} - \tilde{x}^s||^2 \\
&\leq L^2(\mu+1)\mathbf{E}||x_{D(t)}^{s+1} - \tilde{x}^s||^2 \\
&\leq 2L^2(\mu+1)\mathbf{E}||x_{D(t)}^{s+1} - x_t^{s+1}||^2 + 2L^2(\mu+1)\mathbf{E}||x_t^{s+1} - \tilde{x}^s||^2,
\end{aligned} \tag{4}
$$

where the first inequality adopts the Lipschitz smooth property of $f_a(x)$, and the second equality holds because $\delta_x = \frac{||x_{D(t)}^{s+1}||_\infty}{2^{b_x-1}-1}$ and $Q_{(\delta_x,b_x)}(x_{D(t)}^{s+1})$ is an unbiased quantization. The second inequality uses the condition in Step 7 of

Algorithm 1. With similar arguments, we obtain the upper bound of $T_2$ in the following.

$$
\begin{aligned}
T_2 &= \mathbf{E}||\nabla f_a(Q_{(\delta_x,b_x)}(x_{D(t)}^{s+1})) - \nabla f_a(\tilde{x}^s) + \nabla f(\tilde{x}^s) - \nabla f(x_t^{s+1})||^2 \\
&\leq 2\mathbf{E}||\nabla f_a(Q_{(\delta_x,b_x)}(x_{D(t)}^{s+1})) - \nabla f_a(x_{D(t)}^{s+1})||^2 + 2\mathbf{E}||\nabla f_a(x_{D(t)}^{s+1}) - \nabla f_a(\tilde{x}^s) + \nabla f(\tilde{x}^s) - \nabla f(x_t^{s+1})||^2 \\
&\leq 2\mathbf{E}||\nabla f_a(Q_{(\delta_x,b_x)}(x_{D(t)}^{s+1})) - \nabla f_a(x_{D(t)}^{s+1})||^2 + 4\mathbf{E}||\nabla f_a(x_{D(t)}^{s+1}) - \nabla f_a(x_t^{s+1})||^2 \\
&\quad + 4\mathbf{E}||\nabla f_a(x_t^{s+1}) - \nabla f_a(\tilde{x}^s) + \nabla f(\tilde{x}^s) - \nabla f(x_t^{s+1})||^2 \\
&\leq 2\mathbf{E}||\nabla f_a(Q_{(\delta_x,b_x)}(x_{D(t)}^{s+1})) - \nabla f_a(x_{D(t)}^{s+1})||^2 + 4\mathbf{E}||\nabla f_a(x_{D(t)}^{s+1}) - \nabla f_a(x_t^{s+1})||^2 + 4\mathbf{E}||\nabla f_a(x_t^{s+1}) - \nabla f_a(\tilde{x}^s)||^2 \\
&\leq 2L^2\mathbf{E}||Q_{(\delta_x,b_x)}(x_{D(t)}^{s+1}) - x_{D(t)}^{s+1}||^2 + 4L^2\mathbf{E}||x_{D(t)}^{s+1} - x_t^{s+1}||^2 + 4L^2\mathbf{E}||x_t^{s+1} - \tilde{x}^s||^2 \\
&\leq 4L^2(\mu+1)\mathbf{E}||x_{D(t)}^{s+1} - x_t^{s+1}||^2 + 4L^2(\mu+1)\mathbf{E}||x_t^{s+1} - \tilde{x}^s||^2.
\end{aligned}
$$
(5)

where in the third inequality we adopt $\mathbf{E}||\nabla f_a(x_t^{s+1}) - \nabla f_a(\tilde{x}^s) + \nabla f(\tilde{x}^s) - \nabla f(x_t^{s+1})||^2 \leq \mathbf{E}||\nabla f_a(x_t^{s+1}) - \nabla f_a(\tilde{x}^s)||^2$. It is true because $\mathbf{E}||x - \mathbf{E}[x]||^2 \leq \mathbf{E}||x||^2$. The last inequality follows from Step 7 of Algorithm 1. Putting them together, we obtain Lemma 2. $\qquad\square$

***Proof of Theorem* 2.** Define $\bar{x}_{t+1}^{s+1} \triangleq \text{prox}_{\eta h}(x_t^{s+1} - \eta \nabla f(x_t^{s+1}))$. According to equations (8)-(12) in [2], we get

$$
\begin{aligned}
\mathbf{E}&\Big[P(x_{t+1}^{s+1})\Big] \\
&\leq \mathbf{E}\Big[P(x_t^{s+1}) + (L - \frac{1}{2\eta})||\bar{x}_{t+1}^{s+1} - x_t^{s+1}||^2 + (\frac{L}{2} - \frac{1}{2\eta})||x_{t+1}^{s+1} - x_t^{s+1}||^2 - \frac{1}{2\eta}||x_{t+1}^{s+1} - \bar{x}_{t+1}^{s+1}||^2 \\
&\quad + \langle x_{t+1}^{s+1} - \bar{x}_{t+1}^{s+1}, \nabla f(x_t^{s+1}) - u_t^{s+1}\rangle\Big] \\
&\leq \mathbf{E}\Big[P(x_t^{s+1}) + \frac{\eta}{2}||u_t^{s+1} - \nabla f(x_t^{s+1})||^2 + (L - \frac{1}{2\eta})||\bar{x}_{t+1}^{s+1} - x_t^{s+1}||^2 + (\frac{L}{2} - \frac{1}{2\eta})||x_{t+1}^{s+1} - x_t^{s+1}||^2\Big].
\end{aligned}
$$
(6)

Using Lemma 2, we have

$$
\begin{aligned}
\mathbf{E}\Big[P(x_{t+1}^{s+1})\Big] \leq &\mathbf{E}\Big[P(x_t^{s+1}) + \eta L^2(\mu+1)(\Delta+2)||x_{D(t)}^{s+1} - x_t^{s+1}||^2 + \eta L^2(\mu+1)(\Delta+2)||x_t^{s+1} - \tilde{x}^s||^2 \\
&+ (L - \frac{1}{2\eta})||\bar{x}_{t+1}^{s+1} - x_t^{s+1}||^2 + (\frac{L}{2} - \frac{1}{2\eta})||x_{t+1}^{s+1} - x_t^{s+1}||^2\Big].
\end{aligned}
$$
(7)

Define $R_t^{s+1} \triangleq \mathbf{E}\Big[P(x_t^{s+1}) + c_t||x_t^{s+1} - \tilde{x}^s||^2\Big]$, where $\{c_t\}_{t=0}^m$ is a nonnegative decreasing sequence with $c_m = 0$, $c_t = c_{t+1}(1+\beta) + \eta L^2(\mu+1)(\Delta+2)$ and $\beta = \frac{1}{m}$. Therefore,

$$
\begin{aligned}
c_0 &\leq \eta L^2(\mu+1)(\Delta+2) \cdot \frac{(1+\beta)^m - 1}{\beta} \\
&\leq 2\eta L^2(\mu+1)(\Delta+2)m.
\end{aligned}
$$
(8)

From the definition of $R_t^{s+1}$, we obtain

$$
\begin{aligned}
R_{t+1}^{s+1} =&\mathbf{E}\Big[P(x_{t+1}^{s+1}) + c_{t+1}||x_{t+1}^{s+1} - \tilde{x}^s||^2\Big] \\
\leq&\mathbf{E}\Big[P(x_{t+1}^{s+1}) + c_{t+1}(1 + \frac{1}{\beta})||x_{t+1}^{s+1} - x_t^{s+1}||^2 + c_{t+1}(1+\beta)||x_t^{s+1} - \tilde{x}^s||^2\Big] \\
\leq&\mathbf{E}\Big[P(x_t^{s+1}) + c_t||x_t^{s+1} - \tilde{x}^s||^2 + (c_{t+1}(1 + \frac{1}{\beta}) + \frac{L}{2} - \frac{1}{2\eta})||x_{t+1}^{s+1} - x_t^{s+1}||^2 + (L - \frac{1}{2\eta})||\bar{x}_{t+1}^{s+1} - x_t^{s+1}||^2 \\
&+ \eta L^2(\mu+1)(\Delta+2)\tau \sum_{d=D(t)}^{t-1} ||x_{d+1}^{s+1} - x_d^{s+1}||^2\Big].
\end{aligned}
$$
(9)

Summing over $t = 0$ to $m - 1$, we get

$$\sum_{t=0}^{m-1} R_{t+1}^{s+1} \leq \sum_{t=0}^{m-1} R_t^{s+1} + \sum_{t=0}^{m-1} \left[ c_{t+1}(1 + \frac{1}{\beta}) + \frac{L}{2} - \frac{1}{2\eta} + \eta L^2(\mu+1)(\Delta+2)\tau^2 \right] \mathbf{E}||x_{t+1}^{s+1} - x_t^{s+1}||^2$$
$$+ \sum_{t=0}^{m-1} (L - \frac{1}{2\eta}) \mathbf{E}||\bar{x}_{t+1}^{s+1} - x_t^{s+1}||^2. \tag{10}$$

The inequality holds because $\sum_{t=0}^{m-1} \sum_{d=D(t)}^{t-1} ||x_{d+1}^{s+1} - x_d^{s+1}||^2 \leq \tau \sum_{t=0}^{m-1} ||x_{t+1}^{s+1} - x_t^{s+1}||^2.$

Now we derive the bound for $\eta$ to make $\left[ c_{t+1}(1 + \frac{1}{\beta}) + \frac{L}{2} - \frac{1}{2\eta} + \eta L^2(\mu+1)(\Delta+2)\tau^2 \right] \leq 0$. Since $c_t$ is a decreasing sequence, we only need to prove the above inequality for $c_0$. Let $\eta = \frac{\rho}{L}$, where $\rho < \frac{1}{2}$ is a positive constant. After calculations, we obtain the following constraint:

$$8\rho^2 m^2(\mu+1)(\Delta+2) + 2\rho^2(\mu+1)(\Delta+2)\tau^2 + \rho \leq 1. \tag{11}$$

If (11) holds, then

$$\sum_{t=0}^{m-1} R_{t+1}^{s+1} \leq \sum_{t=0}^{m-1} R_t^{s+1} + (L - \frac{1}{2\eta}) \sum_{t=0}^{m-1} \mathbf{E}||\bar{x}_{t+1}^{s+1} - x_t^{s+1}||^2. \tag{12}$$

Because $x_0^{s+1} = \tilde{x}^s$, $x_m^{s+1} = \tilde{x}^{s+1}$ and $c_m = 0$, we have $R(x_0^{s+1}) = P(\tilde{x}^s)$ and $R(x_m^{s+1}) = P(\tilde{x}^{s+1})$. Summing (12) over $s = 0$ to $S - 1$, we get

$$(\frac{1}{2\eta} - L) \sum_{s=0}^{S-1} \sum_{t=0}^{m-1} \mathbf{E}||\bar{x}_{t+1}^{s+1} - x_t^{s+1}||^2 \leq P(x^0) - P(x^*). \tag{13}$$

Using the definition of $G_\eta(x_t^{s+1}) \triangleq \frac{1}{\eta}[x_t^{s+1} - \text{prox}_{\eta h}(x_t^{s+1} - \eta \nabla f(x_t^{s+1}))] = \frac{1}{\eta}(x_t^{s+1} - \bar{x}_{t+1}^{s+1})$, we obtain Theorem 2. $\qquad\square$

## 2 Analysis for Sparse-AsyLPG

**Lemma 3.** *Define $\varphi_t \triangleq \sum_{i=1}^d p_i$. If $\varphi_t \leq \frac{||\alpha_t||_1}{||\alpha_t||_\infty}$, then for $\alpha_t = \sum_{i=1}^d \alpha_{t,i} e_i$, we have $\mathbf{E}||\beta_t||^2 \geq \frac{1}{\varphi_t}||\alpha_t||_1^2$. The equality holds if and only if $p_i = \frac{|\alpha_{t,i}| \cdot \varphi_t}{||\alpha_t||_1}$.*

*Proof Sketch.* From the calculation of $\beta_t$, we obtain $\mathbf{E}||\beta_t||^2 = \sum_{i=1}^d \frac{\alpha_{t,i}^2}{p_i}$. If $\varphi_t \leq \frac{||\alpha_t||_1}{||\alpha_t||_\infty}$, then it can be concluded from Lemma 3 and Theorem 5 in [3] that $\mathbf{E}||\beta_t||^2 \geq \frac{1}{\varphi_t}||\alpha_t||_1^2$, with equality if and only if $p_i = \frac{|\alpha_{t,i}| \cdot \varphi_t}{||\alpha_t||_1}$. $\qquad\square$

**Lemma 4.** *Suppose $\varphi_t \leq \frac{||\alpha_t||_1}{||\alpha_t||_\infty}$, Assumptions 1, 2, 3 hold, and for each $i \in \{1, ..., d\}$, $p_i = \frac{|\alpha_{t,i}| \cdot \varphi_t}{||\alpha_t||_1}$. Denote $\Gamma = \frac{d^2}{4\varphi(2^{b-1}-1)^2} + \frac{d}{\varphi} + 1$, where $\varphi = \min_t\{\varphi_t\}$. Then, for the gradient $u_t^{s+1}$ in Sparse-AsyLPG, we have*

$$\mathbf{E}||u_t^{s+1} - \nabla f(x_t^{s+1})||^2 \leq 2L^2(\mu+1)\Gamma \mathbf{E}\left[ ||x_{D(t)}^{s+1} - x_t^{s+1}||^2 + ||x_t^{s+1} - \tilde{x}^s||^2 \right]. \tag{14}$$

*Proof.*
$$\mathbf{E}||u_t^{s+1} - \nabla f(x_t^{s+1})||^2$$
$$= \mathbf{E}||Q_{(\delta_{\beta_t}, b)}(\beta_t) + \nabla f(\tilde{x}^s) - \nabla f(x_t^{s+1})||^2$$
$$= \mathbf{E}||Q_{(\delta_{\beta_t}, b)}(\beta_t) - \beta_t||^2 + \mathbf{E}||\beta_t + \nabla f(\tilde{x}^s) - \nabla f(x_t^{s+1})||^2$$
$$\leq \frac{d}{4(2^{b-1}-1)^2} \mathbf{E}||\beta_t||^2 + \mathbf{E}||\beta_t - \alpha_t||^2 + \mathbf{E}||\alpha_t + \nabla f(\tilde{x}^s) - \nabla f(x_t^{s+1})||^2$$
$$= \left[ \frac{d}{4(2^{b-1}-1)^2} + 1 \right] \mathbf{E}||\beta_t||^2 - \mathbf{E}||\alpha_t||^2 + \mathbf{E}||\alpha_t + \nabla f(\tilde{x}^s) - \nabla f(x_t^{s+1})||^2 \tag{15}$$

where the second equality holds because $Q_{(\delta_{\beta_t}, b)}(\beta_t)$ is an unbiased quantization. The first inequality uses Lemma 1 and $\mathbf{E}[\beta_t] = \alpha_t$. According to $T_2$ we get

$$
\begin{aligned}
&\mathbf{E}||u_t^{s+1} - \nabla f(x_t^{s+1})||^2 \\
&\leq \left[\frac{d}{4(2^{b-1}-1)^2} + 1\right]\mathbf{E}||\beta_t||^2 - \mathbf{E}||\alpha_t||^2 + 4L^2(\mu+1)\mathbf{E}||x_{D(t)}^{s+1} - x_t^{s+1}||^2 + 4L^2(\mu+1)\mathbf{E}||x_t^{s+1} - \tilde{x}^s||^2.
\end{aligned}
\tag{16}
$$

From Lemma 3, we obtain

$$
\begin{aligned}
&\mathbf{E}||u_t^{s+1} - \nabla f(x_t^{s+1})||^2 \\
&\leq \left[\frac{d^2}{4\varphi(2^{b-1}-1)^2} + \frac{d}{\varphi} - 1\right]\mathbf{E}||\alpha_t||^2 + 4L^2(\mu+1)\mathbf{E}||x_{D(t)}^{s+1} - x_t^{s+1}||^2 + 4L^2(\mu+1)\mathbf{E}||x_t^{s+1} - \tilde{x}^s||^2 \\
&\leq 2L^2(\mu+1)\left[\frac{d^2}{4\varphi(2^{b-1}-1)^2} + \frac{d}{\varphi} + 1\right]\mathbf{E}||x_{D(t)}^{s+1} - x_t^{s+1}||^2 + 2L^2(\mu+1)\left[\frac{d^2}{4\varphi(2^{b-1}-1)^2} + \frac{d}{\varphi} + 1\right]\mathbf{E}||x_t^{s+1} - \tilde{x}^s||^2,
\end{aligned}
\tag{17}
$$

where the first inequality uses $||x||_1 \leq \sqrt{d}||x||_2$ for $x \in \mathbb{R}^d$ and $\mathbf{E}||\beta_t||^2 = \frac{1}{\varphi_t}||\alpha_t||_1^2$ when $p_i = \frac{|\alpha_{t,i}| \cdot \varphi_t}{||\alpha_t||_1}$. The final inequality comes from the upper bound of $T_1$.   □

***Proof sketch of Theorem* 3.** Substituting (14) in (6) and following the proof of Theorem 2, we obtain a convergence rate of

$$
\mathbf{E}||G_\eta(x_{out})||^2 \leq \frac{2L[P(x^0) - P(x^*)]}{\rho(1-2\rho)T},
\tag{18}
$$

if $8\rho^2 m^2(\mu+1)\Gamma + 2\rho^2(\mu+1)\tau^2\Gamma + \rho \leq 1$, where $\Gamma = \frac{d^2}{4\varphi(2^{b-1}-1)^2} + \frac{d}{\varphi} + 1$.   □

# 3   Proof of Theorem 4

The following lemma is a widely used technical result in composite optimization, which is called *3-Point-Property*. Lemma 1 in [1] provides its detailed proofs and extensions.

**Lemma 5.** *If $y_{t+1}^s$ is the optimal solution of*

$$
\min_{y \in \chi} \phi(y) + \frac{1}{2\eta_s}||y - y_t^s||^2,
\tag{19}
$$

*where function $\phi(y)$ is convex over a convex set $\chi$. Then for any $y \in \chi$, we have [1]*

$$
\phi(y) + \frac{1}{2\eta_s}||y - y_t^s||^2 \geq \phi(y_{t+1}^s) + \frac{1}{2\eta_s}||y_{t+1}^s - y_t^s||^2 + \frac{1}{2\eta_s}||y - y_{t+1}^s||^2.
\tag{20}
$$

***Proof of Theorem* 4.** From the update rule of $y_{t+1}^s$, we know that

$$
y_{t+1}^s = \arg\min_y h(y) + \langle u_t^s, y - y_t^s \rangle + \frac{1}{2\eta_s}||y - y_t^s||^2.
\tag{21}
$$

Applying Lemma 5 with $\phi(y) = h(y) + \langle u_t^s, y - y_t^s \rangle$ and $y = x^*$ in (20), we obtain

$$
h(y_{t+1}^s) + \langle u_t^s, y_{t+1}^s - y_t^s \rangle \leq h(x^*) + \langle u_t^s, x^* - y_t^s \rangle + \frac{1}{2\eta_s}||x^* - y_t^s||^2 - \frac{1}{2\eta_s}||x^* - y_{t+1}^s||^2 - \frac{1}{2\eta_s}||y_{t+1}^s - y_t^s||^2.
\tag{22}
$$

Since $f(x)$ is Lipschitz smooth, we have

$$
\begin{aligned}
\mathbf{E}f(x_{t+1}^s) &\leq \mathbf{E}\left(f(x_t^s) + \langle \nabla f(x_t^s), x_{t+1}^s - x_t^s \rangle + \frac{L}{2}||x_{t+1}^s - x_t^s||^2\right) \\
&= \mathbf{E}\left(f(x_t^s) + \theta_s\langle u_t^s, y_{t+1}^s - y_t^s \rangle + \theta_s\langle \nabla f(x_t^s) - u_t^s, y_{t+1}^s - y_t^s \rangle + \frac{L}{2}||x_{t+1}^s - x_t^s||^2\right),
\end{aligned}
\tag{23}
$$

where the first equality uses $x_{t+1}^s - x_t^s = \theta_s(y_{t+1}^s - y_t^s)$. Therefore,

$$
\begin{aligned}
\mathbf{E}P(x_{t+1}^s) =& \mathbf{E}\Big[ f(x_{t+1}^s) + h(x_{t+1}^s) \Big] \\
\leq & \mathbf{E}\Big[ f(x_t^s) + \theta_s\langle u_t^s, y_{t+1}^s - y_t^s\rangle + \theta_s\langle \nabla f(x_t^s) - u_t^s, y_{t+1}^s - y_t^s\rangle + \frac{L}{2}||x_{t+1}^s - x_t^s||^2 + h(x_{t+1}^s) \Big] \\
\leq & \mathbf{E}\Big[ f(x_t^s) + \theta_s\langle u_t^s, y_{t+1}^s - y_t^s\rangle + \theta_s\langle \nabla f(x_t^s) - u_t^s, y_{t+1}^s - y_t^s\rangle + \frac{L}{2}||x_{t+1}^s - x_t^s||^2 + (1 - \theta_s)h(\tilde{x}^{s-1}) + \theta_s h(y_{t+1}^s) \Big] \\
\leq & \mathbf{E}\Big[ \theta_s h(x^*) + \underbrace{\theta_s\langle u_t^s, x^* - y_t^s\rangle}_{T_3} + \frac{\theta_s}{2\eta_s}||x^* - y_t^s||^2 - \frac{\theta_s}{2\eta_s}||x^* - y_{t+1}^s||^2 - \frac{\theta_s}{2\eta_s}||y_{t+1}^s - y_t^s||^2 \\
& + f(x_t^s) + \underbrace{\theta_s\langle \nabla f(x_t^s) - u_t^s, y_{t+1}^s - y_t^s\rangle}_{T_4} + \frac{L}{2}||x_{t+1}^s - x_t^s||^2 + (1 - \theta_s)h(\tilde{x}^{s-1}) \Big],
\end{aligned}
$$

$$(24)$$

where the first inequality uses (23), and the second inequality follows from $x_{t+1}^s = \theta_s y_{t+1}^s + (1 - \theta_s)\tilde{x}^{s-1}$ and the convexity of $h(x)$. We apply (22) in the third inequality. $T_3$ can be bounded as follows.

$$
\begin{aligned}
\mathbf{E}T_3 =& \theta_s\mathbf{E}\langle u_t^s, x^* - y_t^s\rangle \\
=& \mathbf{E}\langle u_t^s, \theta_s x^* + (1 - \theta_s)\tilde{x}^{s-1} - x_t^s\rangle \\
=& \mathbf{E}\langle u_t^s, \theta_s x^* + (1 - \theta_s)\tilde{x}^{s-1} - Q_{(\delta_x,b_x)}(x_{D(t)}^s)\rangle + \mathbf{E}\langle u_t^s, Q_{(\delta_x,b_x)}(x_{D(t)}^s) - x_t^s\rangle \\
=& \mathbf{E}\langle \nabla f_a(Q_{(\delta_x,b_x)}(x_{D(t)}^s)), \theta_s x^* + (1 - \theta_s)\tilde{x}^{s-1} - Q_{(\delta_x,b_x)}(x_{D(t)}^s)\rangle + \mathbf{E}\langle \nabla f_a(Q_{(\delta_x,b_x)}(x_{D(t)}^s)), Q_{(\delta_x,b_x)}(x_{D(t)}^s) - x_t^s\rangle \\
\leq & \mathbf{E}\Big[ f_a(\theta_s x^* + (1 - \theta_s)\tilde{x}^{s-1}) - f_a(Q_{(\delta_x,b_x)}(x_{D(t)}^s)) + f_a(Q_{(\delta_x,b_x)}(x_{D(t)}^s)) - f_a(x_t^s) + \frac{L}{2}||Q_{(\delta_x,b_x)}(x_{D(t)}^s) - x_t^s||^2 \Big] \\
\leq & \mathbf{E}\Big[ \theta_s f(x^*) + (1 - \theta_s)f(\tilde{x}^{s-1}) - f(x_t^s) + \frac{L}{2}||Q_{(\delta_x,b_x)}(x_{D(t)}^s) - x_t^s||^2 \Big],
\end{aligned}
$$

$$(25)$$

where the convexity and Lipschitz smoothness of $f_a(x)$ are adopted in the first inequality. Next we derive the bound of $\mathbf{E}||Q_{(\delta_x,b_x)}(x_{D(t)}^s) - x_t^s||^2$ as follows.

$$
\begin{aligned}
\mathbf{E}||Q_{(\delta_x,b_x)}(x_{D(t)}^s) - x_t^s||^2 &= \mathbf{E}||Q_{(\delta_x,b_x)}(x_{D(t)}^s) - x_{D(t)}^s||^2 + \mathbf{E}||x_{D(t)}^s - x_t^s||^2 \\
&\leq \theta_s\mu\mathbf{E}||x_{D(t)}^s - \tilde{x}^{s-1}||^2 + \mathbf{E}||x_{D(t)}^s - x_t^s||^2 \\
&\leq (1 + 2\theta_s\mu)\mathbf{E}||x_{D(t)}^s - x_t^s||^2 + 2\theta_s^3\mu\mathbf{E}||y_t^s - \tilde{x}^{s-1}||^2 \\
&\leq (1 + 2\theta_s\mu)\mathbf{E}||x_{D(t)}^s - x_t^s||^2 + 2\theta_s^3\mu D.
\end{aligned}
$$

$$(26)$$

where the first equality holds because $Q_{(\delta_x,b_x)}(x_{D(t)}^s)$ is an unbiased quantization and the first inequality comes from Step 7 in Algorithm 3. The second inequality holds because $x_t^s - \tilde{x}^{s-1} = \theta_s(y_t^s - \tilde{x}^{s-1})$. Therefore,

$$
\mathbf{E}T_3 \leq \mathbf{E}\Big[ \theta_s f(x^*) + (1 - \theta_s)f(\tilde{x}^{s-1}) - f(x_t^s) + \frac{(1 + 2\theta_s\mu)L}{2}||x_{D(t)}^s - x_t^s||^2 + \theta_s^3\mu LD \Big]. \tag{27}
$$

Now we bound $T_4$. Define $v_t^s \triangleq \nabla f_a(x_t^s) - \nabla f_a(\tilde{x}^{s-1}) + \nabla f(\tilde{x}^{s-1})$.

$$
\mathbf{E}T_4 = \theta_s\mathbf{E}\langle \nabla f(x_t^s) - u_t^s, y_{t+1}^s - y_t^s\rangle = \underbrace{\theta_s\mathbf{E}\langle \nabla f(x_t^s) - v_t^s, y_{t+1}^s - y_t^s\rangle}_{T_5} + \underbrace{\theta_s\mathbf{E}\langle v_t^s - u_t^s, y_{t+1}^s - y_t^s\rangle}_{T_6}. \tag{28}
$$

$$T_5 = \theta_s \mathbf{E}\langle \nabla f(x_t^s) - v_t^s, y_{t+1}^s - y_t^s\rangle$$

$$\leq \frac{\theta_s}{2\tau L}\mathbf{E}||\nabla f(x_t^s) - v_t^s||^2 + \frac{\tau L\theta_s}{2}\mathbf{E}||y_{t+1}^s - y_t^s||^2$$

$$\leq \frac{\theta_s}{2\tau L}\mathbf{E}||\nabla f_a(x_t^s) - \nabla f_a(\tilde{x}^{s-1})||^2 + \frac{\tau L\theta_s}{2}\mathbf{E}||y_{t+1}^s - y_t^s||^2$$

$$\leq \frac{\theta_s L^2}{2\tau L}\mathbf{E}||x_t^s - \tilde{x}^{s-1}||^2 + \frac{\tau L\theta_s}{2}\mathbf{E}||y_{t+1}^s - y_t^s||^2 \tag{29}$$

$$= \frac{\theta_s^3 L^2}{2\tau L}\mathbf{E}||y_t^s - \tilde{x}^{s-1}||^2 + \frac{\tau L\theta_s}{2}\mathbf{E}||y_{t+1}^s - y_t^s||^2$$

$$\leq \frac{\theta_s^3 LD}{2\tau} + \frac{\tau L\theta_s}{2}\mathbf{E}||y_{t+1}^s - y_t^s||^2,$$

where in the first inequality we use Young's inequality. The second equality follows from $x_t^s - \tilde{x}^{s-1} = \theta_s(y_t^s - \tilde{x}^{s-1})$. Moreover,

$$T_6 = \theta_s \mathbf{E}\langle v_t^s - u_t^s, y_{t+1}^s - y_t^s\rangle$$

$$\leq \frac{\theta_s}{2\tau L}\mathbf{E}||v_t^s - u_t^s||^2 + \frac{\tau L\theta_s}{2}\mathbf{E}||y_{t+1}^s - y_t^s||^2. \tag{30}$$

From the definition of $u_t^s$ and $v_t^s$, we have

$$\mathbf{E}||v_t^s - u_t^s||^2$$

$$= \mathbf{E}||Q_{(\delta_{\alpha_t},b)}(\alpha_t) - \nabla f_a(x_t^s) + \nabla f_a(\tilde{x}^{s-1})||^2$$

$$= \mathbf{E}||Q_{(\delta_{\alpha_t},b)}\Big(\nabla f_a(Q_{(\delta_x,b_x)}(x_{D(t)}^s)) - \nabla f_a(\tilde{x}^{s-1})\Big) - \nabla f_a(Q_{(\delta_x,b_x)}(x_{D(t)}^s)) + \nabla f_a(\tilde{x}^{s-1})||^2$$

$$\quad + \mathbf{E}||\nabla f_a(Q_{(\delta_x,b_x)}(x_{D(t)}^s)) - \nabla f_a(x_t^s)||^2$$

$$\leq \frac{d}{4(2^{b-1}-1)^2}\mathbf{E}||\nabla f_a(Q_{(\delta_x,b_x)}(x_{D(t)}^s)) - \nabla f_a(\tilde{x}^{s-1})||^2 + \mathbf{E}||\nabla f_a(Q_{(\delta_x,b_x)}(x_{D(t)}^s)) - \nabla f_a(x_t^s)||^2$$

$$\leq \frac{dL^2}{4(2^{b-1}-1)^2}\mathbf{E}||Q_{(\delta_x,b_x)}(x_{D(t)}^s) - \tilde{x}^{s-1}||^2 + L^2\mathbf{E}||Q_{(\delta_x,b_x)}(x_{D(t)}^s) - x_t^s||^2$$

$$\leq \Big[\frac{dL^2}{2(2^{b-1}-1)^2} + L^2\Big]\mathbf{E}||Q_{(\delta_x,b_x)}(x_{D(t)}^s) - x_t^s||^2 + \frac{dL^2}{2(2^{b-1}-1)^2}\mathbf{E}||x_t^s - \tilde{x}^{s-1}||^2$$

$$\leq \Big[\frac{dL^2}{(2^{b-1}-1)^2} + 2L^2\Big]\mathbf{E}||Q_{(\delta_x,b_x)}(x_{D(t)}^s) - x_{D(t)}^s||^2 + \Big[\frac{dL^2}{(2^{b-1}-1)^2} + 2L^2\Big]\mathbf{E}||x_{D(t)}^s - x_t^s||^2$$

$$\quad + \frac{dL^2\theta_s^2}{2(2^{b-1}-1)^2}\mathbf{E}||y_t^s - \tilde{x}^{s-1}||^2$$

$$\leq \Big[\frac{dL^2}{(2^{b-1}-1)^2} + 2L^2\Big]\theta_s\mu\mathbf{E}||x_{D(t)}^s - \tilde{x}^{s-1}||^2 + \Big[\frac{dL^2}{(2^{b-1}-1)^2} + 2L^2\Big]\mathbf{E}||x_{D(t)}^s - x_t^s||^2 + \frac{dL^2\theta_s^2 D}{2(2^{b-1}-1)^2}$$

$$\leq \Big[\frac{dL^2}{(2^{b-1}-1)^2} + 2L^2\Big](1 + 2\theta_s\mu)\mathbf{E}||x_{D(t)}^s - x_t^s||^2 + \Big[\frac{2dL^2}{(2^{b-1}-1)^2} + 4L^2\Big]\theta_s\mu\mathbf{E}||x_t^s - \tilde{x}^{s-1}||^2 + \frac{dL^2\theta_s^2 D}{2(2^{b-1}-1)^2}$$

$$\leq (1 + 2\theta_s\mu)L^2\Big[\frac{d}{(2^{b-1}-1)^2} + 2\Big]\mathbf{E}||x_{D(t)}^s - x_t^s||^2 + \theta_s^3 L^2\mu\Big[\frac{2d}{(2^{b-1}-1)^2} + 4\Big]D + \frac{dL^2\theta_s^2 D}{2(2^{b-1}-1)^2},$$

$$\tag{31}$$

where the second equality holds because $Q_{(\delta_{\alpha_t},b)}(\alpha_t)$ is an unbiased quantization, and the first inequality uses Lemma 1. In the fourth and final inequality, we adopt $x_t^s - \tilde{x}^{s-1} = \theta_s(y_t^s - \tilde{x}^{s-1})$. The fifth inequality follows from Step 7 of Algorithm 3. Putting them together, we obtain

$$\mathbf{E}T_4 \leq \tau L\theta_s\mathbf{E}||y_{t+1}^s - y_t^s||^2 + \frac{L\theta_s(1 + 2\theta_s\mu)}{2\tau}\Big[\frac{d}{(2^{b-1}-1)^2} + 2\Big]\mathbf{E}||x_{D(t)}^s - x_t^s||^2 + \frac{LD\theta_s^3}{2\tau}\Big[\frac{d}{2(2^{b-1}-1)^2} + 1\Big]$$

$$+ \frac{\theta_s^4 LD\mu}{2\tau}\Big[\frac{2d}{(2^{b-1}-1)^2} + 4\Big].$$

$$\tag{32}$$

Substituting (32) and (27) in (24), we get

$$
\begin{aligned}
\mathbf{E}P(x_{t+1}^s) \leq & \mathbf{E}\Big[(1-\theta_s)P(\tilde{x}^{s-1}) + \theta_s P(x^*) + \frac{\theta_s}{2\eta_s}(||x^*-y_t^s||^2 - ||x^*-y_{t+1}^s||^2) \\
& + \theta_s^3 \mu LD + \frac{LD\theta_s^3}{2\tau}\Big[\frac{d}{2(2^{b-1}-1)^2}+1\Big] + \frac{\theta_s^4 LD\mu}{2\tau}\Big[\frac{2d}{(2^{b-1}-1)^2}+4\Big] \\
& + \underbrace{\frac{(1+2\theta_s\mu)L}{2}||x_{D(t)}^s - x_t^s||^2 + \tau L\theta_s||y_{t+1}^s - y_t^s||^2 + \frac{L\theta_s(1+2\theta_s\mu)}{2\tau}\Big[\frac{d}{(2^{b-1}-1)^2}+2\Big]||x_{D(t)}^s - x_t^s||^2}_{T_7} \\
& + \underbrace{\frac{L}{2}||x_{t+1}^s - x_t^s||^2 - \frac{\theta_s}{2\eta_s}||y_{t+1}^s - y_t^s||^2}_{T_8}\Big].
\end{aligned}
\tag{33}
$$

Let $\eta_s \theta_s = \frac{1}{\sigma L}$ where $\sigma > 1$, since $\sum_{t=0}^{m-1}\sum_{d=D(t)}^{t-1}||x_{d+1}^s - x_d^s||^2 \leq \tau \sum_{t=0}^{m-1}||x_{t+1}^s - x_t^s||^2$, it can be verified that

$$
\sum_{t=0}^{m-1}(T_7 + T_8) \leq \xi \sum_{t=0}^{m-1}||y_{t+1}^s - y_t^s||^2,
\tag{34}
$$

where $\xi = \tau^2\theta_s^2\Big[\frac{(1+2\theta_s\mu)L}{2} + \frac{(1+2\theta_s\mu)\theta_s L}{2\tau}\big(\frac{d}{(2^{b-1}-1)^2}+2\big)\Big] + \tau L\theta_s + \frac{L\theta_s^2}{2} - \frac{\sigma L\theta_s^2}{2}$. Denote $\Delta = \frac{d}{(2^{b-1}-1)^2} + 2$, if $\tau \leq \frac{\sqrt{\big(\frac{2}{(1+2\theta_s\mu)\theta_s}+\theta_s\Delta\big)^2 + \frac{4(\sigma-1)}{(1+2\theta_s\mu)}} - \big(\frac{2}{(1+2\theta_s\mu)\theta_s}+\theta_s\Delta\big)}{2}$, then $\xi \leq 0$. Suppose the above constraint holds, we have

$$
\sum_{t=0}^{m-1}\mathbf{E}P(x_{t+1}^s) \leq \sum_{t=0}^{m-1}\mathbf{E}\Big[(1-\theta_s)P(\tilde{x}^{s-1}) + \theta_s P(x^*) + \frac{\theta_s}{2\eta_s}(||x^*-y_t^s||^2 - ||x^*-y_{t+1}^s||^2) + \theta_s^3\mu LD + \frac{\theta_s^3 LD\Delta}{4\tau} + \frac{\theta_s^4 LD\Delta\mu}{\tau}\Big].
\tag{35}
$$

Using $\tilde{x}^s = \frac{1}{m}\sum_{t=0}^{m-1}x_{t+1}^s$, we obtain

$$
\mathbf{E}\Big[P(\tilde{x}^s)-P(x^*)\Big] \leq \mathbf{E}\Big[(1-\theta_s)\big(P(\tilde{x}^{s-1})-P(x^*)\big) + \frac{\sigma L\theta_s^2}{2m}(||y_0^s-x^*||^2 - ||y_m^s-x^*||^2) + \theta_s^3\mu LD + \frac{\theta_s^3 LD\Delta}{4\tau} + \frac{\theta_s^4 LD\Delta\mu}{\tau}\Big].
\tag{36}
$$

Dividing both sides of (36) by $\theta_s^2$, summing over $s=1$ to $S$, and using the definition $y_0^s = y_m^{s-1}$ and that $\frac{1-\theta_s}{\theta_s^2} \leq \frac{1}{\theta_{s-1}^2}$ when $\theta_s = \frac{2}{s+2}$, we have

$$
\mathbf{E}\Big[P(\tilde{x}^S)-P(x^*)\Big] \leq \frac{4\Big[P(\tilde{x}^0)-P(x^*)\Big]}{(S+2)^2} + \frac{2\sigma L||\tilde{x}^0-x^*||^2}{m(S+2)^2} + \frac{8\big(\Delta(1+\mu)/\tau + \mu\big)LD\log(S+2)}{(S+2)^2}.
\tag{37}
$$

$\square$