[Reviews · NeurIPS 2019]

Reviewer 1



The paper is well written, and the method seems to be working nicely. The ideas and methodology are good. The main issue is that the experiments are somewhat weak. There are no tests done on common networks, and use a small somewhat "engineered" fully connected network only. Indeed, QSVRG showed similar test problems, but that paper is from 2013. Also, in QSVRG the authors considered two data sets – CIFAR10 and MNIST, and not only MNIST. After rebuttal: the authors have somewhat addressed my concern about the weak experiments. Trying Resnet18 on CIFAR10 is nice, but is considered a toy example. Still, this much improves the quality of the results. In light of this, and the other reviews I upgrade my score to 7.

Reviewer 2



Originality: Algorithms proposed are not completely new, though they are novel combinations of well-known techniques. The theoretical analysis is an important contribution, but it was unclear to me that there were technical breakthroughs instead of mainly building on approaches in prior work. Update after reading author feedback: - Thanks for running / reporting the experiments that I requested. I was a little disappointed that the scalability was underwhelming -- doubling the number of workers from 4 to 8 only improved running time by ~20%. - After also taking into account the other reviewers' comments, I would still support accepting the paper, but at a slightly lower score. Quality: Presentation of multiple algorithms with proper theoretical analyses is pretty much complete. Empirical evaluations could be more extensive. Clarity: Overall the writing was clear, with an easy-to-follow presentation of algorithms that build on top of the previous algorithm. I did get a little lost with some of the notations, but that is understandable for a theory-focused paper. Significance: I rate the overall significance of this paper as high, for demonstrating both theoretically and empirically that asynchrony, quantization, sparsification, and acceleration can be combined together into a single distributed optimization algorithm. My only complaints are with the empirical evaluations. 1. The greatest issue I have is that there is only 1 evaluation / plot (Figure 3b) on total training time. My understanding is that the motivation for all this work is to reduce the training time to epsilon loss, by reducing communication while not impacting convergence rate. With only 1 plot on total training time for 1 run using 1 dataset on 1 model / problem, I am unable to conclude that the proposed algorithms can robustly accomplish the goal of reducing total training time. 2. I also thought that the experiments did not address the question of scalability. Only one setting for number-of-workers was used. I do expect that the proposed algorithms will scale well; nevertheless, I would have expected this evaluation to be done for a paper on *distributed* optimization. 3. Finally, more extension experiments could have been done -- more runs to get error bars, more datasets, more problems, etc. -- to more convincingly establish the robustness of proposed approaches over competitor algorithms.

Reviewer 3



The paper addresses the communication bottleneck in distributed optimization and proposes double quantization for mitigating this. The approach is not particularly novel as many papers have proposed quantization or compression methods. Here are some detailed points: - The technique is not new and indeed there are important references missing. For example, there should be comparison with the paper "signSGD: Compressed Optimisation for Non-Convex Problems" by Bernstein et al. which considers exactly the same problem. - The main theoretical issue is that the results are based on an unbiased quantizer and this is indeed crucial for the proofs. The low-precision quantizer is only unbiased in the domain dom(delta,b). - The motivation behind quantizing from master to workers is less clear. Open-MPI has a broadcast function and the cost is not the same as N unicasts. - The numerical results on 6 servers are quite limited and the achieved gain is not impressive. - To achieve convergence, model parameter quantization should have a vanishing error which means less and less quantization as iterations increase. What happens to total number of bits then for reaching a desired accuracy? The relationship of parameter mu and number of bits is unclear. To conclude, after the rebuttal I am more convinced about the contributions of this paper. I still find the theoretical novelty of the work incremental; however, the algorithmic novelty and numerical results are quite good, so I increase my score to 6.

[Author Response · NeurIPS 2019]

We thank all reviewers for their time and effort in reviewing our paper.

**– Reviewer** 1 **–**

We set up experiments on PyTorch with ResNet18 (He et al., 2016) on CIFAR10 (Krizhevsky, 2009). The model
size is about 44MB. We use 50k training samples and 10k evaluation samples. For direct comparison, no data
augmentation is used. The batch size is 128. The learning rate starts from 0.1, and is divided by 10 at 150 and
250 epochs. We set $b_x = 8$, $b = 4$ for low-precision algorithms. The sparsity budget $\varphi_t = ||\alpha_t||_1/||\alpha_t||_\infty$ and
$\theta_s = 2/(s+2)$, $\eta_s = lr/\theta_s$ (parameters in Acc-AsyLPG). In Figure 1, we plot the training loss and test accuracy
w.r.t. epochs, and provide the total transmitted bits until the training loss first gets below 0.17. It shows that our
algorithms achieve similar accuracy and effectively reduce the communication cost compared to benchmarks.

Figure 1: Evaluations on CIFAR10: training loss (1st column), test accuracy (2nd column) and total number of transmitted bits.

**– Reviewer** 2 **–**

**(Running time)** The statistics of running time in Figure 3(b) in our paper
are obtained by averaging results of 5 runs in order to make the evalua-
tions accurate. In Figure 2 here, we provide the total running time of lo-
gistic regression on rcv1 and a 3-layer fully connected neural network on
MNIST. The experimental settings are the same as Section 5. The statistics
in both graphs are recorded until the training loss first gets below 0.5. The
results show that our algorithms can effectively reduce the total running time.
**(Scalability)** We present the running time on MNIST using 4, 8, 12 workers in
Figure 3 here. The experimental settings are the same as Section 5.2. Each bar
represents the total running time which is decomposed into communication (top,
light, include transmission, encoding and decoding) and computation (bottom,
dark), and is recorded until the training loss first gets below 0.1. The results show
that algorithm speedup increases in the number of workers. More evaluations of
training ResNet18 (model size 44MB) on CIFAR10 are shown in Figure 1 above.
We will release our code on GitHub in the final version.

Figure 2: Decomposition of time consumption. Top: rcv1. Bottom: MNIST.

**– Reviewer** 3 **–**

**(Comparison with existing results)** (Bernstein et al., 2018) studies bi-direction 1-bit compression between master
and workers. In their case, the master and workers exchange quantized gradients, whereas in our case, the master
receives quantized gradients from workers and sends quantized model vectors to them. This difference leads to a very
different analysis. The key novel components of our work, compared to existing results (including signSGD), include
the following. (i) We propose the new double quantization scheme (DB). The gradient quantizer, though unbiased,
is nontrivial to analyze, because it is evaluated on quantized model vectors. Since the function $f$ is nonlinear, the
stochastic gradients are biased. As a result, our algorithms cannot be analyzed with arguments used in full-precision
distributed SGD analysis, and require new proofs. (ii) We further integrate sparsification and momentum into DB, and
establish convergence rates under asynchrony. We will be sure to include more related references in the final version,
e.g., (Wang et al., 2017), (Jiang & Agrawal, 2018), (Chen et al., 2018) and (Tang et al., 2019).

**(Quantizer)** Note that with our selections of $\delta_x$, $\delta_{\alpha_t}$ and $\delta_{\beta_t}$, the unquantized
coordinates of $x_{D(t)}$, $\alpha_t$, and $\beta_t$ all lie in the convex hull of the corresponding
domains $\mathrm{dom}(\delta, b)$. In this case, the quantizer is unbiased. Such a quantizer is
equivalent to that in QSVRG (Alistarh, 2018; Yu et al., AISTATS, 2019: Section 4.1)
(also see Lemma 1 in Supplementary for details). Note that we can also adopt other
biased **model quantizer** such as clipping, as long as the precision loss satisfies Eq.

Figure 3: Scalability test on MNIST.

(2) or (4). Similar results can be proven with minor modifications of our analysis. Also note that although a 1-to-$N$
broadcast is cheaper than $N$ 1-to-1 unicasts, broadcasting a quantized vector is still much more communication efficient
than broadcasting a full-precision vector.

**(Scalability)** See Fig. 1,2,3 here for more evaluations on scalability and other datasets/models.

**(Accuracy of model quantizer)** $\mu$ is a hyperparameter to control the precision loss. When $\mu$
is fixed, we choose $b_x$ to satisfy Eq. (2). In Figure 4 here, we set $\mu = 0.5$ and study how the
accuracy of model quantizer improves with iterations when running AsyLPG on MNIST. We
see that the quantization error diminishes. Thus, the number of transmitted bits increases as the

Figure 4: The accuracy of model quantizer.

number of iteration grows. Table 1 in manuscript records the total number of bits for reaching the desired accuracy,
which validates the communication efficiency of our algorithms compared to benchmarks. Moreover, Table 1 evaluates
the total transmitted bits under different $\mu$ for attaining the same accuracy.

[Meta-Review · NeurIPS 2019]

The paper combines model and gradient compression, which is an interesting and relevant topic. It combines these aspects with asynchronous SGD updates and momentum. While reviewers uniformly liked the main contributions, they also agreed that the current literature overview is insufficient, and that scaling experiments are not impressive enough in terms of time savings from 4->8 nodes and were only presented for small networks so far. This was partially addressed in the rebuttal. We strongly encourage the authors to improve related work and the other issues mentioned in reviews and in the rebuttal phase. Additional relevant work for example includes https://arxiv.org/abs/1905.10936 (appearing simultaneously), and https://arxiv.org/abs/1901.09847 , and the line of work around https://epubs.siam.org/doi/pdf/10.1137/18M1166134 and the references therein.